# The D165H Polymorphism of QiMYB-like-1 Is Linked to Interactions between Tannin Accumulation, Herbivory and Biogeographical Determinants of *Quercus ilex*

**DOI:** 10.3390/ijms24010151

**Published:** 2022-12-21

**Authors:** Alejandro Gallardo, David Morcuende, Manuela Rodríguez-Romero, María Isabel Igeño, Fernando Pulido, Alberto Quesada

**Affiliations:** 1Departamento de Bioquímica, Facultad de Veterinaria, Universidad de Extremadura, Avenida De La Universidad, 10003 Cáceres, Spain; 2IPROCAR Research Institute, 10003 Cáceres, Spain; 3TECAL Research Group, University of Extremadura, Avenida De La Universidad, 10003 Cáceres, Spain; 4Indehesa Research Institute, Universidad de Extremadura, Avenida Virgen del Puerto 2, 10600 Plasencia, Spain; 5Dirección General de Política Forestal, Junta de Extremadura, Avenida Luis Ramallo, 06800 Mérida, Spain; 6InBio G+C Research Institute, University of Extremadura, Avenida De La Universidad, 10003 Cáceres, Spain

**Keywords:** QiMYB-like-1, *Quercus ilex*, phenolics, (condensed) tannins, herbivory

## Abstract

The accumulation in the leaves and young stems of phenolic compounds, such as hydrolyzable and condensed tannins, constitutes a defense mechanism of plants against herbivores. Among other stressing factors, chronic herbivory endangers *Quercus ilex*, a tree playing a central role in Mediterranean forests. This work addressed the connections between the chemical defenses of *Q. ilex* leaves and their susceptibility to herbivory, quantitative traits whose relationships are modulated by environmental and genetic factors that could be useful as molecular markers for the selection of plants with improved fitness. A search for natural variants detected the polymorphism D165H in the effector domain of QiMYB-like-1, a TT2-like transcription factor whose family includes members that control the late steps of condensed tannins biosynthesis in different plant species. QiMYB-like-1 D165H polymorphism was screened by PCR-RFLP in trees from six national parks in Spain where *Q. ilex* has a relevant presence, revealing that, unlike most regions that match the Hardy-Weinberg equilibrium, homozygous plants are over-represented in “Monfragüe” and “Cabañeros”, among the best examples to represent the continental Mediterranean (cM) ecosystem. Accordingly, the averages of two stress-related quantitative traits measured in leaves, herbivory index and accumulation of condensed tannins, showed asymmetric distributions depending on the clustering of trees based on ecological and genetic factors. Thus, the impact of herbivory was greater in managed forests with a low density of trees from the cM region, among which QiMYB-like-1 D165 homozygotes stand out, whereas condensed tannins accumulation was higher in leaves of QiMYB-like-1 H165 homozygotes from low-density forests, mainly in the Pyrenean (Py) region. Besides, the correlation between the contents of condensed tannins and total tannins vanished after clustering by the same factors: the cM region singularity, forest tree density, and QiMYB-like-1 genotype, among which homozygous shared the lowest link. The biogeographical and genetic constraints that modulate the contribution of condensed tannins to chemical defenses also mediated their interactions with the herbivory index, which was found positively correlated with total phenolics or tannins, suggesting an induction signal by this biotic stress. In contrast, a negative correlation was observed with condensed tannins after tree clustering by genetics factors where associations between tannins were lost. Therefore, condensed tannins might protect *Q. ilex* from defoliation in parks belonging to the cM ecosystem and carrying genetic factor(s) linked to the QiMYB-like-1 D165H polymorphism.

## 1. Introduction

Seven national parks in the Iberian Peninsula share a significant presence of *Quercus ilex*, with a continuous distribution at the Center-South and with a more scattered distribution in the North ([1,2], Appendix A). Holm oaks are suffering oak decline, a severe disease that endangers many *Quercus* species [3]. The selection of *Q. ilex* trees resistant to biotic stress is critical to improving the sustainability of the Mediterranean forest.

Tannins are a broad spectrum of phenolic compounds that play a role as chemical defenses against biotic stress in plants [4]. These molecules are accumulated in the leaves of *Quercus* species to protect themselves from herbivore damage by inhibition of digestive proteases [5]. Two classes of tannins are synthesized in plants: hydrolyzable tannins that are derived from the common precursor 3-dehydroshikimate, and proanthocyanidins or condensed tannins (CT), which are flavonoid molecules produced from phenylpropanoids [6]. Gene expression for late enzymes that synthesize condensed tannins is regulated by a tripartite complex of transcription factors including an R2-R3 MYB factor, a basic helix-loop-helix protein, and a WD40 protein, respectively named TT2, TT8, and TTG1, evoking the *transparent testa* phenotypes conferred by mutations in the flavonoid pathway of *Arabidopsis thaliana* [6]. Depending on plant species, between 100-200 paralogs encode the R2-R3 type MYB-family, whose members regulate development processes and the responses to biotic and abiotic stresses by controlling multiple steps of metabolic processes, mainly the pathway for flavonoid biosynthesis but also to lignin and other cell wall components [7]. Accordingly, QsMYB1, whose transcript was detected in the external bark of *Quercus suber* [8] and accumulates specifically under cork-producing conditions [9], including heat and drought stresses and plant recovery [10], binds to promoter regions of genes involved in lignin and suberin biosynthesis [11].

Biotic stress by herbivory has been shown to produce signals that regulate tannin biosynthesis [12] and, reciprocally, defoliation by insects was found to be modulated by the genetic background and positively correlated with phenolic content of leaves from *Quercus robur*, indicating a compensatory feeding response [13]. Furthermore, defoliation of *Q. ilex* increases the gene expression of its group III shikimate dehydrogenase isoenzyme (SDH2) [14], the role of which has been assigned to the synthesis of gallic acid from 3-dehydroshikimate in *Vitis vinifera* [15]. Therefore, the only known enzyme involved in the biosynthesis of hydrolyzable tannins in plants could be induced by the attack of herbivores, although, nevertheless, the simultaneous infection by the oomycete that causes the decline of holm oak in Mediterranean forests, *Phytophthora cinnamomi*, blocks this response, probably worsening plant health when subjected to this double biotic stress [14].

The oak genome database makes available the chromosomal and coding sequences of *Q. robur* [16] and provides information to analyze genes of closely related species such as *Q. ilex*, which sequence has been recently described by preprint although not yet been made available (doi.org/10.1101/2022.10.09.511480, accessed on 6 December 2022). This work addressed the identification of TT2-like protein polymorphism in *Q. ilex*, its geographic distribution pattern, and possible association with phenolics traits and/or herbivore susceptibility.

## 2. Results

### 2.1. Polymorphism of QiMYB-like-1, a Putative Component of the MBW Complex in Q. ilex

The coding sequences for QiMYB-like-1 to QiMyb-like-4 polypeptides were identified in the oak genome and PCR-amplified and sequenced from *Q. ilex* (Figure 1). A search of polymorphisms in the four DNA fragments detected two closely linked variants, G526C and A529G, in the coding sequence for QiMYB-like-1, and PCR-MboI-RFLP revealed the QiMYB-like-1 D165H alleles and corresponding genotypes (Figure 2).

**Figure 1 ijms-24-00151-f001:**
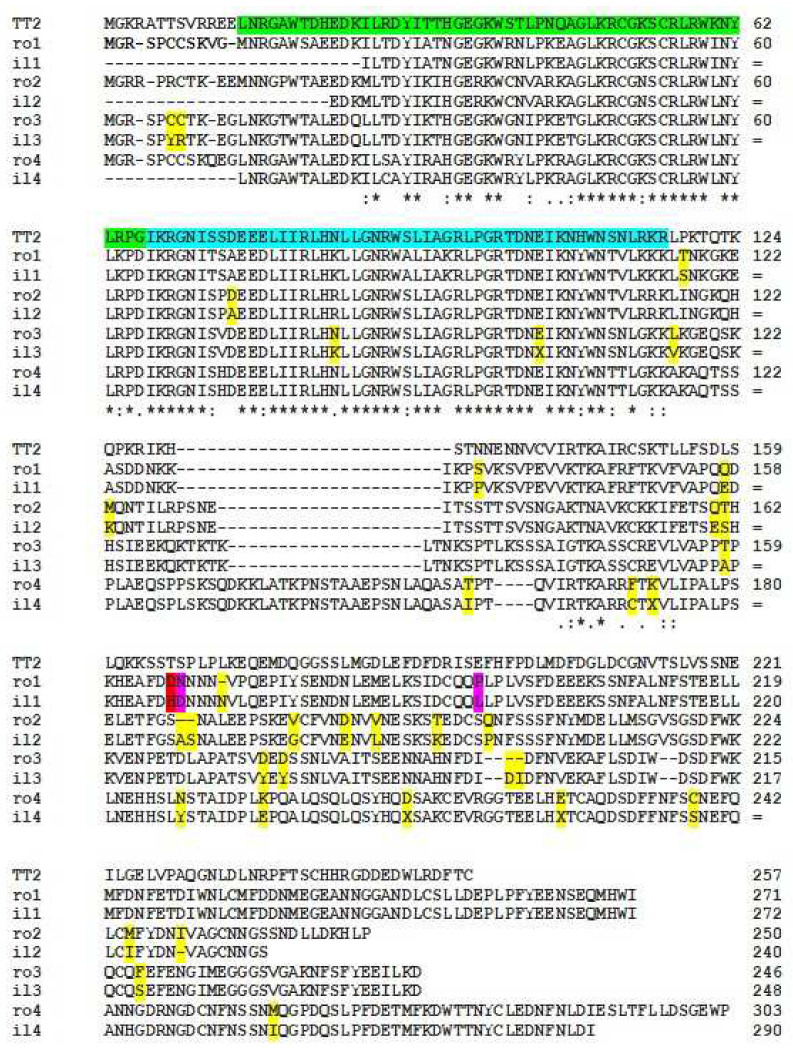
Comparative analysis of Quercus MYB-like proteins. Blastp identified four homologs (Qrob_P0344270.2, Qrob_P0660350.2, Qrob_P0304670.2, and Qrob_P0304630.2) in the *Q. robus* genome assembly PMN1 (haploid, Pseudomolecule + unassigned scaffolds; urgi.versailles.inra.fr/blast/, accessed on 6 December 2022) by using the TT2 sequence from *A. thaliana* (Q9FJA2) and, after the design of specific primers (Table 1) and PCR amplification, the four homologs from *Q. ilex* were identified and sequenced (QiMYB-like-1 D165 allele, QBI90131; QiMYB-like-1 H165 allele, QBI90132; QiMYB-like-2, QBI90133; QiMYB-like-3, QBI90134; QiMYB-like-4, QBI90135). Green/blue, predicted R2/R3 repeats ([6] and references therein); yellow, polymorphisms between *Q. robur* and *Q. ilex* sequences; pink+red, polymorphisms between *Q. robur* and *Q. ilex* sequences and also between QiMYB-like-1 and QiMYB-like-2 haplotypes; red, polymorphism D165H detected by RFLP (Figure 2). Clustal symbolizes identity (*), strong conservation (:) or weak conservation (.) among aligned protein sequences.

### 2.2. Geographical Distribution of the QiMYB-like-1 Genotypes

The occurrence of genotypes carrying QiMYB-like-1 alleles was analyzed in holm oaks from six national parks and found in Hardy-Weinberg equilibrium in all but one case (Table 2). A higher ratio of homozygous plants than expected in trees from Monfragüe National Park promoted the detected deviation in the HW equilibrium. Although Cabañeros was not statistically significant, a similar tendency against heterozygous in this locus was found in the other region representing the cM biogeographical context, a common ecosystem widely spread in the southern half of the Iberian Peninsula [1,2].

### 2.3. Non-Random Distribution of Quantitative Traits Related to Chemical Defenses Depends on Biogeographic, Ecologic, or Genetic Factors

The leaves of *Q. ilex* sampled in the six national parks accumulated significant amounts of the three families of molecules determined as representatives of holm oak chemical defenses: TP, TT, and CT (Figure 3). The comparison of TP averages among national parks evidences similar values, whereas the TT contents of the trees in the Pyrenean parks were the lowest and CT accumulation presents a decreasing gradient from northern to southern regions. Besides, the highest susceptibility to herbivores was found in Cabañeros, which presented in turn one of the lowest CT contents. When contrasted according to the regional singularity Py vs cM, their differences in susceptibility to defoliation and between their contents in CT and TT are maximized (Figure 3). On the other hand, none of the averages of the quantitative variables differed according to the density of the forest or the distribution in the genotypes defined for QiMYB-like-1.

Regional (cM vs. Py), habitat type (LDL vs. HDL), and genetic (QiMYB-like-1 D165H polymorphism) interactions depending on the distribution of quantitative traits related to biotic stress were only evident for CT content and herbivory index, while TT content only showed highly significant variation (*p* < 0.001) with respect to regional grouping (Table 3). Thus, we observed that herbivory susceptibility varied by region (*p* < 0.001) and its interaction with genotype (G × R, *p* < 0.027)) and with density (R × D, *p* < 0.001), although not by all three simultaneously. Similarly, the CT content shows significant differences at the regional (*p* < 0.001), ecological (*p* < 0.003), and genetic (*p* < 0.033) levels, individually and according to the GxD interaction (*p* < 0.028), although also not for all the three variables simultaneously (Table 3). Furthermore, QiMYB-like-1 H165 homozygous (HH) trees grown in low-density forests accumulate higher amounts of CT, while plants from the cM region are more prone to attack by herbivores when located in low-density forests or sharing the QiMYB-like-1 homozygous D165 genotype (DD; Figure 4).

### 2.4. Susceptibility to Defoliation and/or Chemical Defenses Depends on Biogeographic, Ecological, or Genetic Factors

Accumulation of the three quantitative traits indicative of the phenolic defenses analyzed in this work (TP, TT, and CT) were found to be correlated with each other in populations of *Q. ilex* from the Iberian Peninsula, in agreement with the overlap between these families of molecules (Table 4). Best fits were found within every National Park, where regression coefficients reached above or near 0.8 for TP and TT, although slopes between TP and CT decreased to 0.4–0.5 from northern to southern parks. In contrast, TT and CT contents were the less correlated chemical defenses suggesting that CT contribution to total tannins and/or phenolics is more variable in holm oaks when latitude decreases in the Iberian Peninsula (Table 4) [1,2]. Moreover, after classifying the plants according to the three singularities defined as independent variables for this study, the contribution of CT loses the link with TP or TT accumulations in the cM region. Similarly, QiMYB-like-1 homozygous plants had TT and TP or CT and TP contents more closely correlated with each other than CT and TT, whose association was completely lost after grouping by biogeographical criteria, as also occurs with classification based on forest density (Table 4). On the other hand, the associations between chemical defenses and the defoliation index presented a variable sign, being mostly positive for TP and TT, and negative for CT, although the latter relationship is strongly dependent on the cM singularity (Table 5), the same factor that dissociates the CT and TT contents (Table 4). Again, regional singularity dominates over ecological or genetic variables, since negative correlations between herbivory index and CT content are only found for QiMYB-like-1 homozygous or plants grown on low-density forests in the cM national parks, where these tannin molecules seem to protect leaves from insect attack.

## 3. Discussion

Leaves from *Q. ilex* trees grown in six national parks of the Iberian Peninsula [1,2] have been analyzed in search of possible links among genetic, ecologic, and biogeographical determinants of biotic stress. Results shown in this work evidence that these factors might modulate interactions between herbivore susceptibility and the accumulation of chemical defenses, represented in holm oaks by phenolic compounds, mainly tannins and particularly condensed tannins [17].

Genomic sequences encoding four TT2-like polypeptides, the R2R3-MYB component of the MBW tripartite complex in *A. thaliana* [6], were detected in a preliminary version of the oak genome database (Figure 1) [16]. The four R2R3-MYB proteins presented polymorphisms between the two *Quercus* species, *Q. robur* and *Q. ilex*, with most variants located in the C-terminus, a poorly conserved and acidic-rich domain with an effector function to regulate mRNA expression from the target gene (s), whereas the two predicted DNA binding motifs (HTH1 and HTH2) at the N-terminus are less prone to inter-species evolution (Figure 1). D165H, a polymorphism detected in the effector domain of QiMYB-like-1, presents a biased distribution of genotypes in the continental-Mediterranean (cM) region (Table 2), a biogeographical context dominated by Mediterranean forests that share almost all environmental factors (latitude, altitude, temperature, and raining), suggesting a genetic selection by biogeographical constraint (s). After this clustering, the bias from HW equilibrium reaches a maximum only for cM, where a skewed distribution of the QiMYB-like-1 polymorphism adds genetic support to its singularity.

The two quantitative characters that represent chemical defenses in the leaves of *Q. ilex*, TT, and CT, as well as the defoliation index that is directly related to biotic stress, are differentially biased towards the biogeographic singularities cM and Py (Figure 3): defoliation and TT content are lower and CT accumulation is higher in Py than in cM national parks. The two other independent variables utilized in this work, the ecological constraints involving forest density, and the genetic factor(s) linked to QiMYB-like-1 polymorphism, do not affect the distributions of averages, although ANOVA detects interactions among them on susceptibility to defoliation and CT contents (Table 3), two out of the three quantitative traits related to biological stress whose averages presented asymmetric distributions (Figure 3). However, different QiMYB-like-1 genotypes are associated with both, defoliation and CT accumulation (Figure 4), explaining why the three-way ANOVA for the defoliation index is just above the significance threshold based on biogeographic, forest density, and QiMYB-like-1 polymorphism singularities (Table 3). Interestingly, QiMYB-like-1 homozygotes are the overrepresented genotypes in the cM biogeographical singularity, where DD plants have the lowest defoliation rate whilst the HH genotype shares the highest CT content in low-density forests, which suggests that at least one genetic determinant for each of these biotic stress indices might be linked to the QiMYB-like-1 locus.

The analysis of the regression fits between plant parameters related to biotic stress complements the comparison of their averages, confirming the impact of biogeographical, ecological, and genetic singularities on the quantitative characters of *Q. ilex*. Thus, all chemical defenses were found positively correlated among themselves, although clustering by biogeographic singularities clearly breaks the associations between CT and TT in the cM region, probably indicating that contribution of the former is distinct and more variable in this environment (Table 4). This effect dominates over ecological or genetic singularities since classification by forest density or QiMYB-like-1 genotype has little or no effect unless the plants considered are from the cM region. The correlation analysis between the herbivory index and chemical defenses showed an asymmetric distribution of the averages of these variables, although additional associations were detected, like the positive link between TP and defoliation rates under almost any condition used for plant classification (Table 5). Positive correlations were also found between TT and defoliation susceptibility when the total of plants is considered, although it is lost when we grouped according to the biogeographic region except for Py, LDF, and DH heterozygotes. Although the upregulation of one trait by the other cannot be excluded, this phenomenon could also be explained if both the TP and TT chemical defenses and the rate of herbivory responded to the same environmental signals. In contrast, defoliation rates and CT contents in the cM singularity were found negatively correlated (Table 5), consistently with the biased distributions of the same variables (Figure 4) and indicating a protective effect of CT in plants associated with biogeographic, ecological, and genetic determinants.

## 4. Materials and Methods

### 4.1. Plant Material

*Quercus ilex* subsp. *rotundifolia* was sampled from six national parks in the Iberian Peninsula with a significant presence of holm oaks [1,2]; Appendix A. Ordered from highest to lowest latitude, these parks were: Aigüestortes, Cabañeros, Guadarrama, Monfragüe, Ordesa, and Sierra Nevada. Among them, four parks represent the cM and Py geo-climatic areas mentioned in this work. Cabañeros and Monfragüe national parks (continental-Mediterranean, cM) are characterized by mild, short winters and long and very hot summers, with altitudes over 500 m, 16 °C average annual temperature, and irregular rainfall around 500 mm. In contrast, Aigües Tortes and Ordesa (Pyrenean, Py) present cold and long winters and mild summers, altitude over 1300 m, 8 °C average temperature, and over 1000 mm rainfall. The two other national parks screened, Guadarrama and Sierra Nevada, present mixed conditions. Sierra Nevada is a high mountain park, from 900 to 3400 m, with the highest peaks in the Iberian Peninsula and an average temperature of 4 °C. On the contrary, the proximity to the Mediterranean Sea provides strong insolation and relative drought, with around 700 mm of rainfall. Guadarrama, also a high mountain (from 900 m), has a less Mediterranean influence, with very cold winters and mild summers, average temperatures below 7 °C, and rainfall of 1200 mm.

### 4.2. Experimental Design, Treatments, and Tissue Processing

The healthy leaves (dark green color, regardless of traces of insect attack) were collected in June 2015, when their development is finished and stabilized, taking them from branches located around 2 m above the ground, avoiding those that are excessively shaded and/or accessible to herbivorous mammals. From the six national parks, two habitats were screened (Appendix A). First, the “dehesa” or low-density forest (LDF, hereafter in this work) corresponds to the Mediterranean agroforestry system characterized by scattered oak trees with undergrowth grazed by livestock, whereas the second habitat is represented by dense or closed forest formations of oaks and undergrowth cover in a high proportion (>40 percent) of the ground (high-density forest, HDF). An average of 100 leaves were collected for each of the 360 trees sampled according to the following scheme: six national parks, two habitats per park, three locations per habitat, and 10 adult plants were selected to represent the entire range of ages per location. Leaves were stored at −80 °C until processed for analysis of their defoliation index (fraction of defoliated leaves per tree), accumulation of chemical defenses, and DNA polymorphism. Defoliation was visually estimated in all leaves collected, and the percentage of leaves with any damage compatible with herbivory by insects per every tree was registered.

### 4.3. Chemical Analyses of Phenolics

Frozen leaf tissues were hand-ground using a mortar and pestle in liquid nitrogen. Fifty milligrams of milled tissue were extracted with 250 μL 50% (*v*/*v*) aqueous methanol for 60 min in an ultrasonic bath. The crude extracts were centrifuged at 10,000 rpm for 5 min at 4 °C and the supernatant was collected and stored at −80 °C.

Total phenolic (TP) content was determined by the Folin–Ciocalteu method according to [18] and following the modifications by [14]. Briefly, 200 µL of appropriately diluted extract (1:50) were mixed with 1 mL of 10% Folin-Ciocalteu reagent (Sigma-Aldrich, Madrid, Spain) followed by 800 µL of 7.5% (*w*/*v*) Na_2_CO_3_. After 40 min incubation in the dark at room temperature, absorbance was measured at 725 nm. Gallic acid (Sigma-Aldrich, Madrid, Spain) was the reference standard, and the results were expressed as mg of gallic acid equivalents (G.A.E.) per g dry leaf material.

Total tannin (TT) content was determined by the radial diffusion method in the presence of BSA [17] and following the modifications by [19]. Gel Petri dishes were prepared using a solution of 50 mM acetic acid and 60 μM ascorbic acid and adjusting to pH 5. One percent agarose (Sigma-Aldrich, Madrid, Spain) was added to the solution and boiled (heated) until homogenized. After cooling to 42 °C, 0.1% bovine serum albumin (Sigma Aldrich, Madrid, Spain) was added and stirred until it dissolves. Uniform wells were made on plates containing the same volume of solidified agarose and constant volumes (10 μL) of samples were loaded, incubated at 30 °C for 120 h, and the sizes of the halos of precipitation were measured and transformed into TT by means of extrapolation in the corresponding calibration, which was performed with tannic acid (Sigma Aldrich, Madrid, Spain) and results were expressed as mg of tannic acid equivalents (T.A.E.) per g of dry leaf material.

Condensed tannins (CT) concentration was measured according to the method described by [20] and following the modifications by [14]. Crude extracts were mixed thoroughly with 100 volumes of n-butanol/acetone 1:1 (46% each) plus HCL (1.85%) and ferric ammonium sulfate (4%). The mixture was heated at 70 °C for 45 min and, after being cooled, the absorbance at 550 nm was measured. A calibration was performed using procyanidin B2 as reference (Sigma-Aldrich, Madrid, Spain), and results were expressed as mg of procyanidin B equivalents (PB.E.) per g dry leaf material.

### 4.4. Purification of Nucleic Acids

Frozen leaf tissues were hand-ground using a mortar and pestle in liquid nitrogen. DNA was isolated from 100 mg of tissue using an i-genomic Plant DNA Extraction Mini kit (Intron Biotechnology, Gyeonggi, Korea), according to the manufacturer’s protocol. Nucleic acids were analyzed and quantified by spectrophotometry (260 nm/280 nm), and the integrity of the preparations was further evaluated by agarose gel electrophoresis.

### 4.5. PCR and RFLP Analyses

Four TT2-like coding sequences were amplified from *Q. ilex* by using specific primers (Table 1) in a Prime G Thermal Cycler (Techne, supplied by Thermo Fisher Scientific, Waltham, MA, USA). The amplified PCR products were used for MboI (Invitrogen, supplied by Thermo Fisher Scientific, massachusetts, USA) digestion, which was performed using a 20 μL mixture containing 18 μL purified PCR product (MEGAquick-spin kit, Intron Biotechnology, Gyeonggi, Korea) and one unit of restriction endonuclease in the appropriate reaction buffer. The incubation temperature was set according to the manufacturer’s instructions for the restriction endonuclease. The resulting restriction fragments were separated on 1.2% agarose gels stained with SYBR-safe DNA gel stain (Invitrogen, supplied by Thermo Fisher Scientific, Waltham, MA, USA). The genotyping of plants by PCR-RFLP was performed by comparison of fragment sizes with the molecular weight marker Gene Ruler DNA Ladder 100 bp Plus (Thermo Fisher Scientific, Waltham, MA, USA) loaded in each agarose gel.

### 4.6. Statistical Analysis

The effects of the biogeographical region (cM vs. Py), forest type (LDF vs. HDF), and genotype on the quantitative traits used to measure biological stress were analyzed through analysis of variance (two-way ANOVA). The region, forest type, and genotype were used as random factors whereas the content of total phenolics (TP), content of total tannins (TT), content of condensed tannins (CT), and defoliation (in%) were dependent variables. The relationships among the 4 quantitative traits representing plant chemical defenses (TP, TT, CT) and susceptibility to herbivory (defoliation) were analyzed by using linear Pearson correlation.

## 5. Conclusions

One main finding of this work is that distinct phenolic compounds that accumulated in the leaves of *Q. ilex* might interact differentially on susceptibility to insect herbivory. Thus, TP and TT do not seem to play significant protection against this biotic stress, since the three traits are parallelly induced. However, the negative correlation of CT content and susceptibility to herbivory suggests that, in the cM biogeographic region that best represents the agroforestry ecosystem known as “dehesa”, the trees that grow in low-density forests (LDF) and are QiMYB-like-1 D165 or H165 homozygotes would present a higher content of CT and a lower susceptibility to herbivory. This fact could explain the anomalously high frequency found of these genotypes in the best preserved “dehesas” in Spain and could be useful to improve the management of their oak forests.

## Figures and Tables

**Figure 2 ijms-24-00151-f002:**
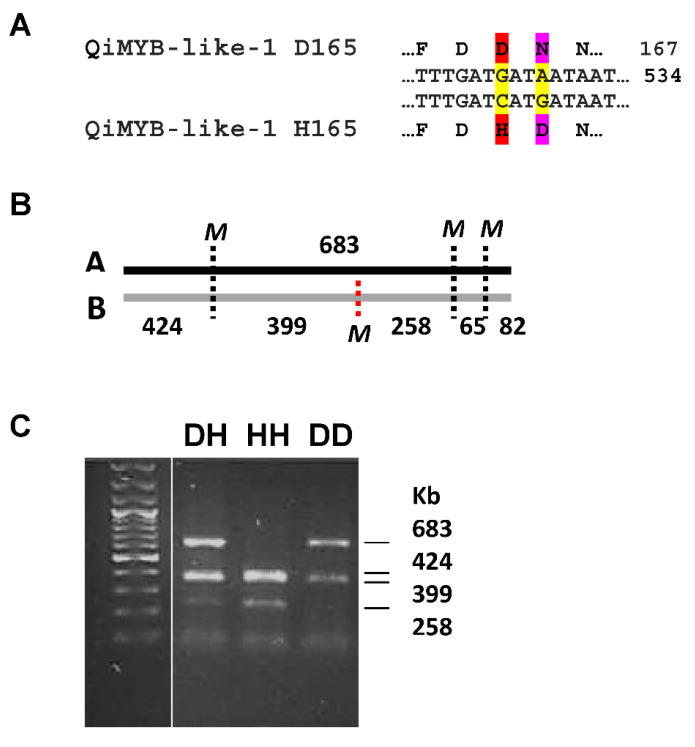
PCR-RFLP detection of QiMYB-like-1 genotypes. (**A**) QiMYB-like-1 D165H polymorphism, coding sequence encoded peptides. (**B**) MboI restriction map of PCR fragments amplified from *Q. ilex* by primers QIMYB-like-1 F and R, shown in Table 1. (**C**) Resolution in agarose gel (1.5%) electrophoresis (0.5 × TBE) of MboI fragments from QIMYB-like-1 specific PCR. Yellow, DNA polymorphisms; red/pink, protein polymorphisms; red, polymorphism detected by RFLP.

**Figure 3 ijms-24-00151-f003:**
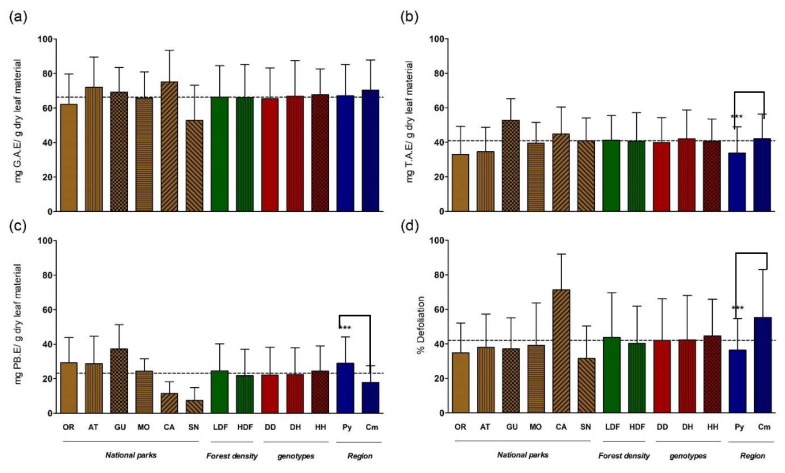
Constitutive levels of: (**a**), total phenolics or TP; (**b**), total tannins or TT; (**c**), condensed tannins or CT; and (**d**), percentage of defoliation in leaves of Quercus ilex. Trees were grouped by their singularities based on: biogeography, Pyrenean (PY) and continental-Mediterranean (cM); forest density, high-density forest (HDF) and low-density forest (LDF); genotypes QUEil; MYB-like-1 D165 homozygous (DD), QUEil;MYB-like-1 H165 homozygous (HH), and QUEil;MYB-like-1 D/H165 heterozygous (DH). The bars in the graph represent the mean plus standard error. The dashed line indicates the mean value of the measured samples. The significance of the differences was statistically assessed by using a two-tailed alternative *t*-test. *** *p* < 0.001 (confidence level 99.9%).

**Figure 4 ijms-24-00151-f004:**
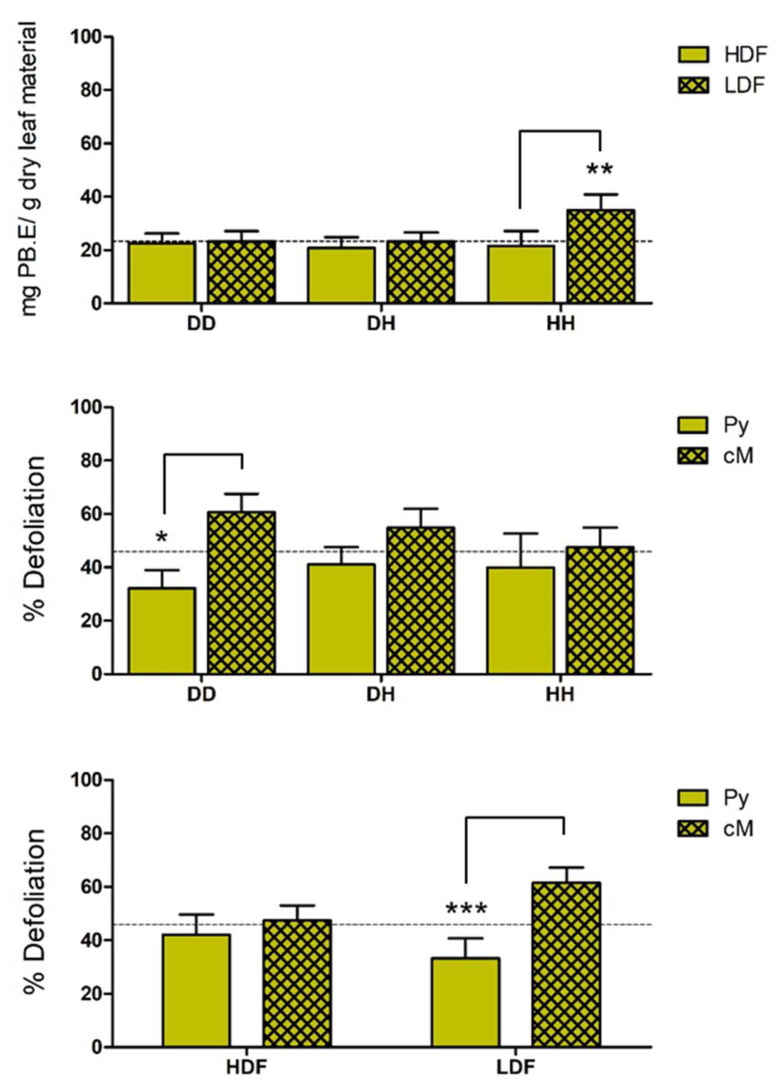
Results from ANOVA and pairwise comparisons between CT and herbivore susceptibility depending on plant singularities (like in Figure 3). The bar graph represents the mean and standard error for a confidence interval of 95%. The dashed line indicates the mean value of the measured samples. Confidence levels were assessed by two-way ANOVA: * 95%, ** 99%, *** 99.9%.

**Table 1 ijms-24-00151-t001:** Primers used in this work.

Coding Sequence	Name	Sequence 5’–3’	Amplicon ^a^
QIMYB-like-1	p1-F	ATGGGCAGAAGTCCTTGC	1229
	p1-R	TGCTTCATGGGGTGTCAAT	
QIMYB-like-2	p2-F	TCAAAATGGGTAGGCGTCCT	1059
	p2-R	ATTCATCGCTTCACTGCATC	
QIMYB-like-3	p3-F	AATCAAACACCCAATTTAAGATCAG	1145
	p3-R	TGTCAGCTTGGTTAGTTGGC	
QIMYB-like-4	p4-F	GAGAGAGAGAAATGGGGAGGA	1209
	p4-R	GGTGTGTAGTGCGTACGTGGA	

^a^ Amplicon size (bp) from genomic DNA.

**Table 2 ijms-24-00151-t002:** Geographical distribution of QiMYB-like-1 genotypes.

		Observed (Expected) Genotypes ^3^				
NP ^1^	N ^2^	DD	DH	HH	JI-Square ^4^	*p*-Value ^5^
OR	52	22 (20.9)	22 (24.1)	8 (6.9)	0.4	0.53
AT	52	23 (24.2)	25 (22.5)	4 (5.2)	0.63	0.43
GU	52	22 (20.3)	21 (24.4)	9 (7.3)	0.99	0.318
MO	59	16 (11)	19 (29)	24 (19)	6.97	0.0082 **
CA	60	25 (21.6)	22 (28.8)	13 (9.6)	3.34	0.067
SN	51	24 (24.7)	23 (21.6)	4 (4.7)	0.22	0.64
Py	104	45 (45.1)	47 (46.8)	12 (12.1)	0.0026	0.96
cM	119	41 (31.8)	41 (59.4)	37 (27.8)	11.45	0.0007 ***

^1^ Analyzed trees from populations of all national parks (NP): OR, “Ordesa”; AT, “AigüesTortes”; GU, “Guadarrama”; MO, “Monfragüe”; CA, “Cabañeros”; SN, “Sierra Nevada”, Py, Pyrenean forests (OR + AT); cM, continental-Mediterranean forests (MO + CA). ^2^ Number of trees genotyped in every population. ^3^ QiMYB-like-1 genotypes result from the combination of QiMYB-like-1 D165 (D) and QiMYB-like-1 H165 (H) alleles, described in Figure 1. ^4^ χ^2^-test (degrees of freedom = 1), contrasted with reference values, appropriate *p*-values ^5^ and confidence levels: ≥3.84, 95% (**); ≥6.64, 99% (***); ≥10.83, 99.9%.

**Table 3 ijms-24-00151-t003:** Interaction effects (two-way ANOVA) of genotype, region, and forest density on chemical defense production and herbivore susceptibility.

	Cases ^1^	SS ^2^	df	MS ^3^	F	*p*
TT	G	229.155	2	114.577	0.519	0.596
	R	3012.700	1	3012.700	13.640	<0.001 ***
	D	25.687	1	25.687	0.116	0.733
	G × R	365.170	2	182.585	0.827	0.439
	G × D	373.856	2	186.928	0.846	0.430
	R × D	47.668	1	47.668	0.216	0.643
	G × R × D	54.166	2	27.083	0.123	0.885
	Residual	46,604.937	211	220.876		
CT	G	1031.73	2	515.87	3.462	0.033 *
	R	7321.21	1	7321.21	49.129	<0.001 ***
	D	1346.37	1	1346.37	9.035	0.003 **
	G × R	125.65	2	62.82	0.422	0.657
	G × D	1079.67	2	539.84	3.623	0.028 *
	R × D	84.63	1	84.63	0.568	0.452
	G × R × D	466.89	2	233.44	1.567	0.211
	Residual	31,443.03	211	149.02		
Herbivory	G	449.4	2	224.7	0.453	0.636
	R	12,129.6	1	12,129.6	24.475	<0.001 ***
	D	298.5	1	298.5	0.602	0.439
	G × R	3634.3	2	1817.2	3.667	0.027 *
	G × D	1210.1	2	605.0	1.221	0.297
	R × D	5750.7	1	5750.7	11.604	<0.001 ***
	G × R × D	3013.5	2	1506.8	3.040	0.050
	Residual	104,570.7	211	495.6		

^1^ G, Genotypes DD, DH and HH (freedom degrees, 2); R, regions (biogeography) cM and Py (freedom degrees, 1); D, forest densities HDF and LDF (freedom degrees, 1), as defined in Figure 3. ^2^ Sum of squares, Type III. ^3^ Mean square. Confidence levels for associations detected: 95% (*); 99% (**); 99.9% (***).

**Table 4 ijms-24-00151-t004:** Linear regression ^1^ between tree defenses: effects of biogeographical, ecological, and genetic singularities.

	C ^2^	N ^3^		TP		TT	
				r	*p*	r	*p*
National Parks	OR	60	TT	0.778	<0.001 ***		
			CT	0.717	<0.001 ***	0.457	<0.001 ***
	AT	60	TT	0.855	<0.001 ***		
			CT	0.664	<0.001 ***	0.430	<0.001 ***
	GU	60	TT	0.857	<0.001 ***		
			CT	0.676	<0.001 ***	0.428	<0.001 ***
	MO	60	TT	0.861	<0.001 ***		
			CT	0.497	<0.001 ***	0.334	0.009 **
	CA	60	TT	0.858	<0.001 ***		
			CT	0.470	<0.001 ***	0.334	0.009 **
	SN	60	TT	0.919	<0.001 ***		
			CT	0.462	<0.001 ***	0.375	0.004 **
Total of plants	NP	360	TT	0.741	<0.001 ***		
			CT	0.475	<0.001 ***	0.287	<0.001 ***
	LDF	180	TT	0.727	<0.001 ***		
			CT	0.488	<0.001 ***	0.254	<0.001 ***
	HDF	180	TT	0.753	<0.001 ***		
			CT	0.464	<0.001 ***	0.319	<0.001 ***
	DD	142	TT	0.684	<0.001 ***		
			CT	0.475	<0.001 ***	0.220	0.012 *
	DH	132	TT	0.792	<0.001 ***		
			CT	0.478	<0.001 ***	0.322	<0.001 ***
	HH	62	TT	0.708	<0.001 ***		
			CT	0.384	0.002 **	0.258	0.043 *
Py/cM singularity	Py	120	TT	0.796	<0.001 ***		
			CT	0.652	<0.001 ***	0.438	<0.001 ***
	cM	120	TT	0.864	<0.001 ***		
			CT	0.153	0.095	0.105	0.253
	LDF	120	TT	0.800	<0.001 ***		
			CT	0.404	<0.001 ***	0.149	0.105
	HDF	120	TT	0.832	<0.001 ***		
			CT	0.359	<0.001 ***	0.164	0.075
	DD	86	TT	0.745	<0.001 ***		
			CT	0.302	0.005 **	0.022	0.841
	DH	88	TT	0.863	<0.001 ***		
			CT	0.460	<0.001 ***	0.284	0.007 **
	HH	49	TT	0.784	<0.001 ***		
			CT	0.278	0.053	0.108	0.462

^1^ Pearson coefficients (r) and associated *p* are shown. ^2^ Clustering of trees, according to: Upper section, national parks where samples were taken (above section, region names like in Table 2); Central section, total sampling (NP) or grouped by forest density LDF (low density forests) and HDF (high density forests) and genetic (QiMYB-like-1 alleles as described in Table 2); Lower section, only trees from the Py and cM region are considered for correlation analysis.^3^ Sample size. Confidence levels for associations detected: 95% (*); 99% (**); 99.9% (***).

**Table 5 ijms-24-00151-t005:** Linear regression ^1^ between herbivory and defenses: effects of biogeographical, ecological, and genetic singularities.

	C ^2^		TP		TT		CT	
		N ^3^	r	*p*	r	*p*	r	*p*
Total of plants	NP	360	0.266	<0.001 ***	0.134	0.011 *	−0.067	0.209
	LDF	180	0.366	<0.001 ***	0.234	0.002 **	−0.058	0.439
	HDF	180	0.154	0.039 *	0.032 *	0.674	−0.092	0.222
	DD	142	0.276	0.001 **	0.127	0.148	−0.069	0.435
	DH	132	0.268	0.002 **	0.134	0.125	−0.037	0.671
	HH	62	0.262	0.039 *	0.224	0.080	−0.188	0.143
Py/cM singularity	Py	120	0.363	<0.001 ***	0.207	0.024 *	0.232	0.011 *
	cM	120	0.184	0.044 *	0.112	0.223	−0.333	<0.001 ***
	LDF	120	0.378	<0.001 ***	0.328	<0.001 ***	−0.217	0.017 *
	HDF	120	0.134	0.144	0.126	0.171	−0.177	0.054
	Py + LDF	60	0.479	<0.001 ***	0.256	0.050	0.426	<0.001 ***
	Py + HDF	60	0.268	0.038 *	0.191	0.144	0.093	0.486
	cM + LDF	60	0.355	0.005 **	0.190	0.146	−0.433	<0.001 ***
	cM + HDF	60	0.003	0.979	0.007	0.959	−0.378	0.003 **
	DD	86	0.231	0.032 *	0.190	0.079	−0.277	0.01 *
	DH	88	0.267	0.012 *	0.244	0.022 *	−0.123	0.254
	HH	49	0.112	0.445	0.170	0.242	−0.297	0.038 *

^1^ Pearson coefficients (r) and associated *p* are shown. ^2,3^ Similar to Table 4. Confidence levels for associations detected: 95% (*); 99% (**); 99.9% (***).

## Data Availability

The data presented in this study are available on request from the corresponding author.

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
