# Peer review of "The D165H Polymorphism of QiMYB-like-1 Is Linked to Interactions between Tannin Accumulation, Herbivory and Biogeographical Determinants of *Quercus ilex"

_ijms, 2022, doi:10.3390/ijms24010151_

Round 1

Reviewer 1 Report

For those interested in Quercus research, but not familiar with specific genes, it will be advisable to put the known name of the gene or that indicative of the function, e.g., transcription factor, enzyme, etc.

The genome of Q. ilex has been recently released. Published as preprint in BioRxiv.

To me, the abstract is a little bit long, and I would recommend reduce it.

While possible, shorten the sentences. Long sentences are difficult to be understood.

Can you provide some characteristics of the surveyed trees? Estimated age, canopy, acorn production, acorn morphotype, or whatever you consider of relevance. Also, a photo of the trees can be provided as supplementary material. Include a figure with a map of the locations.

What are the climate conditions at Sierra Nevada?

The TT content analysis is not clearly presented. How to quantify TT?

The factorial ANOVA is not in the statistics section.

From my experience leaves, depending on the position, use to have a different profile, which is also related to the developmental stages. Not all the leaves are similar.

The leaf sampling is not clear, at least to me. Which is the biological replicate? How many leaves from each tree? What does defoliation visual estimation mean?

“… percentage of leaves with any damage compatible with herbivory by insects was registered.”. I do not understand; of the total sample, of the total tree, …? You have indicated that healthy leaves were collected.

To my experience the variability in total phenolic compounds in leaves of the same tree is very high, so is not a very useful parameter. It depends on the developmental stage of the leaf, position in the canopy, age of the tree.

To me the conclusion is not supported by the data presented. Data and hypothesis should be validated by analysing gene expression products, either of the TF or key enzymes of the phenolic pathway

Author Response

We appreciate the comments made by the reviewers on this work. Below we explain the changes made to the manuscript in response to all the points made, including the numbering of the affected lines and their text highlighted in yellow, in the last submitted version of the manuscript.

For those interested in Quercus research, but not familiar with specific genes, it will be advisable to put the known name of the gene or that indicative of the function, e.g., transcription factor, enzyme, etc.

Answer (lines 77-79): the name of the transcription factors that make up the tripartite complex is now explained in the introduction section.

The genome of Q. ilex has been recently released. Published as preprint in BioRxiv.

Answer (lines 103-105): this preprint is now mentioned in the introduction section and its DOI is referenced.

To me, the abstract is a little bit long, and I would recommend reduce it.

Answer (lines 24-58): the length of the abstract has been reduced by 5% (from 400 to 381 words).

While possible, shorten the sentences. Long sentences are difficult to be understood.

Answer: after reviewing the entire manuscript, the length of the longest sentences has been reduced to less than or equal to five lines.

Can you provide some characteristics of the surveyed trees? Estimated age, canopy, acorn production, acorn morphotype, or whatever you consider of relevance. Also, a photo of the trees can be provided as supplementary material. Include a figure with a map of the locations.

Answer (lines 132-144): based on this comment we have extended, in the materials and methods section, the explanation about the sampling carried out. Initially we did not consider it relevant for this study to record the characteristics of the trees. Instead, we sampled each location (6 national parks x 2 habitats x 3 locations x 10 plants = 360 trees tested), and as now explained, “…and 10 adult plants were selected to represent all the range of ages by locality”.

Second answer to this comment (lines 1-16, supplementary material at the end of the manuscript, in a separated section): in addition, and as also recommended to us, we now include two additional figures as supplementary material. The first is a map with the location of the holm oak forests, meadows and natural parks analyzed in this work. The second consists of a photographic composition showing the two analyzed habitats, low- and high-density forests, in two representative national parks of the Pyrenean and continental-Mediterranean regions, Ordesa and Cabañeros, respectively.

What are the climate conditions at Sierra Nevada?

Answer (lines 122-129): the climatic conditions of the national parks of Sierra Nevada and also of Guadarrama are now provided in the material and methods section.

The TT content analysis is not clearly presented. How to quantify TT?

Answer (lines 172-175): The procedure to quantify TT has been explained in somewhat more detail in the material and methods section.

The factorial ANOVA is not in the statistics section.

Answer (line 212): Added the term "two-way" to ANOVA, referring to the factorial ANOVA that was performed.

From my experience leaves, depending on the position, use to have a different profile, which is also related to the developmental stages. Not all the leaves are similar.

Answer (lines 132-135): The first sentence of the materials and methods section "Experimental design, treatments and tissue processing" has been modified to include additional details of the sampling performed. Thus, it is now explained that leaf samples were taken in similar physiological and developmental states.

The leaf sampling is not clear, at least to me. Which is the biological replicate? How many leaves from each tree? What does defoliation visual estimation mean?

Answer (lines 141-144): as explained above, we have expanded the explanation on the sampling performed in the materials and methods section, mentioning that 100 leaves per tree were analyzed (susceptibility to defoliation and chemical content). In addition, the last sentence of the paragraph (lines 147-149) explains the procedure for measuring the defoliation index, which is expressed as the percentage of leaves defoliated by insects for each sampled tree.

“… percentage of leaves with any damage compatible with herbivory by insects was registered.”. I do not understand; of the total sample, of the total tree, …? You have indicated that healthy leaves were collected.

Answer (lines 132 and 148-149); as explained above, we have expanded the explanation on the sampling performed in the materials and methods section, mentioning that “…healthy leaves (dark green color, regardless of traces of insect attack) were collected”, and that “…the percentage of leaves with any damage compatible with herbivory by insects per every tree was registered”.

To my experience the variability in total phenolic compounds in leaves of the same tree is very high, so is not a very useful parameter. It depends on the developmental stage of the leaf, position in the canopy, age of the tree.

Answer (lines 132-135): as explained above, the first sentence of the materials and methods section "Experimental design, treatments and tissue processing" has been modified to include additional details of the sampling performed. Thus, it is now explained that leaf samples were taken in similar physiological and developmental states, in order to reduce variability in chemical content.

To me the conclusion is not supported by the data presented. Data and hypothesis should be validated by analyzing gene expression products, either of the TF or key enzymes of the phenolic pathway

Answer (lines 375-377): As suggested, the following sentence was added to the last paragraph of the conclusions “…, that should be confirmed in future work by analyzing gene expression products for target enzymes and/or regulators involved in the tannin biosynthesis,…”. In addition, the sections of the manuscript where the paragraphs indicate the main conclusions of this work (last sentences of the abstract, introduction, and discussion) have been carefully screened, ensuring that conditional verb forms are always used.

Reviewer 2 Report

This manuscript was very interesting for revision, and the results are significant, since Q. ilex is playing a central role in Mediterranean forests, and it is a relict species. The introduction could be improved, and authors could write something about the background of this research. I suggest separating Conclusions from Discussion and giving some recommendations.  Minor suggestions are commented on in the text. 

Author Response

We appreciate the comments made by the reviewers on this work. Below we explain the changes made to the manuscript in response to all the points made, including the numbering of the affected lines and their text highlighted in yellow, in the last submitted version of the manuscript.

This manuscript was very interesting for revision, and the results are significant, since Q. ilex is playing a central role in Mediterranean forests, and it is a relict species. The introduction could be improved, and authors could write something about the background of this research.

Answer (lines 92-100): A new paragraph has been added to the introduction section, giving information about previous results and a new reference [15].

I suggest separating Conclusions from Discussion and giving some recommendations. 

Answer (line 366-380): the conclusions section has been added, and the text moved from the last paragraph of the discussion of the previous version.

Minor suggestions are commented on in the text. 

Answer: all minor points have been addressed in the last version of the manuscript.

Reviewer 3 Report

The topic of the paper „The D165H polymorphism of QiMYB-like-1 is linked to interactions between tannin accumulation, herbivory and biogeographical determinants of Quercus ilex” is novelty and very interesting for readers.

This work focused on the connections between chemical defenses of Q. ilex leaves and their susceptibility to herbivory, quantitative traits whose relationships are modulated by environmental and genetic factors that could be useful as molecular markers for the selection of plants with improved fitness.

Authors concluded that, condensed tannins might protect Q. ilex from defoliation in parks belonging to the continental-Mediterranean ecosystem, represented by the Monfragüe and Cabañeros regions, and genetic factor(s) linked to the QiMYB-like-1 D165H polymorphism.

Therefore, distinct phenolic compounds that accumulated in the leaves of Q. ilex interact differentially on insect herbivory.

The manuscript is well written, and the text is easy to read.

The design research is appropriate.

The results are consistent and clearly presented.

Author Response

We appreciate the comments made by the reviewers on this work.

Round 2

Reviewer 1 Report

Nice work. Congratulations-